# Interrelated Mechanism by Which the Methide Quinone Celastrol, Obtained from the Roots of *Tripterygium wilfordii*, Inhibits Main Protease 3CL^pro^ of COVID-19 and Acts as Superoxide Radical Scavenger

**DOI:** 10.3390/ijms21239266

**Published:** 2020-12-04

**Authors:** Francesco Caruso, Manrose Singh, Stuart Belli, Molly Berinato, Miriam Rossi

**Affiliations:** Department of Chemistry, Vassar College, Poughkeepsie, NY 12604, USA; manrosesingh@gmail.com (M.S.); Belli@vassar.edu (S.B.); mberinato@vassar.edu (M.B.)

**Keywords:** celastrol, methide quinone, COVID-19, thiolate, protease, cyclic voltammetry

## Abstract

We describe the potential anti coronavirus disease 2019 (COVID-19) action of the methide quinone inhibitor, celastrol. The related methide quinone dexamethasone is, so far, among COVID-19 medications perhaps the most effective drug for patients with severe symptoms. We observe a parallel redox biology behavior between the antioxidant action of celastrol when scavenging the superoxide radical, and the adduct formation of celastrol with the main COVID-19 protease. The related molecular mechanism is envisioned using molecular mechanics and dynamics calculations. It proposes a covalent bond between the S(Cys145) amino acid thiolate and the celastrol A ring, assisted by proton transfers by His164 and His41 amino acids, and a π interaction from Met49 to the celastrol B ring. Specifically, celastrol possesses two moieties that are able to independently scavenge the superoxide radical: the carboxylic framework located at ring E, and the methide-quinone ring A. The latter captures the superoxide electron, releasing molecular oxygen, and is the feature of interest that correlates with the mechanism of COVID-19 inhibition. This unusual scavenging of the superoxide radical is described using density functional theory (DFT) methods, and is supported experimentally by cyclic voltammetry and X-ray diffraction.

## 1. Introduction

Human use of plants as sources of medicinal benefit predates written history, and there is a growing interest in traditional plant-based medicines. Celastrol is a methide quinone triterpene isolated from the roots of *Tripterygium wilfordii,* or “God of Thunder” vine. It has been used in traditional Chinese medicine for hundreds of years [1] to treat chronic inflammations, autoimmune conditions, neurodegenerative diseases, and cancer-related symptoms [2,3,4]. Toxicity concerns may limit celastrol administration as a drug. In a specific toxicity test, different doses of celastrol were orally administered to mice [5] and showed no significant changes. However, side effects of celastrol administration have been reported, for instance, cardiotoxicity upon chronic treatment [6], and infertility [7]. To overcome celastrol solubility and pharmacokinetic issues, several methodologies have been tested, such as exosomes [8], lipid nanospheres [9], nanoencapsulation [10], liposomes [11,12], polymeric micelles [13,14], sugar-silica nanoparticles [15], and a self-microemulsifying drug delivery system [16]. For instance, celastrol-loaded mesoporous silica nanoparticles that are sugar-decorated have shown increased specific anticancer activity with no induced toxicity in HeLa and A549 cells [15]. Celastrol is also implicated in the NF-κB pathway [17] by interacting with the IKK kinases in a dose-dependent manner. Thus, celastrol likely contributes to its anti-inflammatory and anti-tumor activities by inhibiting NF-κB activation possibly through targeting Cys-179 in IKK-β [18]. Indeed, celastrol interactions with thiol groups have already been described in the literature: (1) celastrol can react with protein thiols in human cervical HeLa cells in a unique covalent and reversible manner [19]. (2) Its quinone methide structure can react specifically with the thiol groups of cysteine residues, forming covalent protein adducts [20]. (3) It shows thiol-related effects on the human monocytic leukemia cell line U937 proliferation [21]. (4) The cytotoxic effect of ionizing radiation in vitro is enhanced with celastrol administration, and its quinone methide moiety is essential for this radiosensitization. Celastrol induced the thiol reactivity and inhibited the activities of antioxidant molecules, such as thioredoxin reductase and glutathione [22]. In addition, reactive oxygen species production by ionizing radiation was augmented. (5) Celastrol promotes proteotoxic stress, supported by the induction of heat-shock proteins, HSP72, through a thiol-dependent mechanism; these findings imply that celastrol targets proteostasis by disrupting sulfyhydryl homeostasis in human glioblastoma cells [23]. (6) In addition, it was seen that celastrol reduced lipopolysaccharides (LPS)-induced expression of inflammatory cytokines, such as tumor necrosis factor (TNF)-α, interleukin (IL)-6, IL-12, and IL-1β. These inhibitory effects of celastrol on LPS were reversed by thiol donors (N-acetyl-L-cysteine and dithiothreitol), suggesting that the thiol reactivity of celastrol contributes to its inhibitory effects on macrophages. These results provide a novel mechanism of action by which celastrol contributes to the anti-inflammatory activity of *T. wilfordii* [24]. This is of interest, since inflammatory symptoms are present in coronavirus disease 2019 (COVID-19) patients, including an unusual multisystem inflammatory syndrome in children (MIS-C). (7) Celastrol’s biological effects, including inhibition of glucocorticoid receptor activity, can be blocked by the addition of excess free thiol, suggesting a chemical mechanism whereby this natural product could modify key reactive thiols [25].

The interaction between cysteine and quinones has been noted [26] and includes a recent description of the quinone embelin establishing an important covalent bond with Cys145 of the main COVID-19 protease 3CL^pro^ to explain the inhibitory mechanism [27]. Since the methide quinone celastrol shows inhibition towards SARS-CoV 3CL^pro^ [28], such an association between celastrol and the active site cysteine in the COVID-19 protease is supported. Moreover, celastrol antiviral activity is described for infectious bronchitis virus [29], influenza A [30], hepatitis C [31], dengue [32], and HIV [33].

Indeed, our described quinone embelin inhibition mechanism on 3CL^pro^ implicates Cys145 assisted through H-bonds from nearby amino acids, and strongly resembles the mechanism of embelin antioxidant activity toward the superoxide radical [34,35]. Both of these two chemical reactions underscore quinone electron affinity. The superoxide transfers its unpaired electron to the quinone embelin through a π–π interaction [34,35], while in the main protease the Cys145 thiolate is also π attracted by the embelin quinone centroid, as seen in docking results. This driving force contributes to the formation of a covalent bond between S(thiolate) and an embelin positively charged carbonyl moiety [27]. Specifically, the Cys145-His41 diad, conserved in all versions of SARS viruses, provides the perfect arrangement for cleavage of the (Cys145) S-H bond assisted by the N-imidazole(His41) acceptor in the embelin case. Finally, among the most effective repurposed drugs against COVID-19 is the corticosteroid dexamethasone [36], which is a methide quinone (as celastrol), and which acts in a similar way as embelin [27]. Its mechanism of action on the main protease 3CL^pro^ is also similar to methyl prednisolone [37], another corticosteroid methide quinone used against COVID-19 [38]. Therefore, the association of celastrol anti-SARS inhibition activity, IC_50_ of 10.3 µM [28], with a similar related mechanism seems apparent. Here, we describe the mechanism of celastrol, Scheme 1, at the active site of the main COVID-19 protease, 3CL^pro^, using crystal structure, electrovoltaic antioxidant measurements, docking, and density functional theory (DFT) methods.

## 2. Results and Discussion

### 2.1. X-ray Diffraction

Beautiful yellow crystals of celastrol were obtained after solvent evaporation in chloroform solution. These crystals were isostructural to an acetonitrile solvated crystal structure of celastrol [39,40]. The asymmetric unit contains two celastrol molecules along with a solvent chloroform molecule. The two molecules form a striking elliptical dimeric structure through two pairs of strong hydrogen bonds at either end of the celastrol molecule. The solvent molecule lies outside of this dimer (see Figure 1). The H-bond values at one end of the dimer are O1-H1···O104 2.842(6) Å and angle 145.8°, and O103-H103···O2 2.607(6) Å and angle 177°, while at the other end of the dimer they are O3-H3···O102 2.573(5) Å and angle 169°, and O101-H101···O4 2.948(5) Å and angle 151º. Each of the two molecules also shows an intramolecular hydrogen bond between the carbonyl oxygen and the adjacent hydroxyl group: O1-H1···O2 2.741(7) Å and angle 113°, and O101-H101···O102 2.707(6) Å and angle 115°. The two molecules do not differ very much, and a superposition of them shows little variation. The three saturated rings are in a chair conformation. Ring B, adjacent to the planar methide quinone ring, is almost coplanar in an envelope conformation with the substituted C6 atom being 0.25 Å out of the plane as seen in Figure 2.

The chloroform solvent provides for additional intermolecular interactions in the crystal: a hydrogen bond C1S-H1S···O4 (donor-acceptor 3.234(9) Å) and C-H-O bond angle 141° along with a halogen bond between Cl2 and the carbonyl O102 of celastrol at a distance of 3.088 Å, shorter than the sum of van der Waals radii, and with the C-Cl-O angle of 155°, Figure 3. Table 1 shows crystal data.

### 2.2. Electrochemistry

The antioxidant capability of celastrol towards the superoxide radical was studied using a variation of classic cyclovoltammetry with a rotating ring disk electrode (RRDE) method recently developed by our lab [41]. The superoxide radical is generated in situ in a voltaic cell using anhydrous dimethyl sulfoxide (DMSO) as solvent, through bubbling of a controlled amount of oxygen. The superoxide radical is obtained at enough negative potential so that O_2_ captures an electron from the working electrode
O_2_ + e^−^ → O_2_^−•^(1)

The configuration of the voltaic cell includes, besides the working rotating disk electrode, a reference electrode and a ring electrode around the disk. The ring potential is chosen positive enough for oxidation of the superoxide so that the reaction opposite to **(1)** is performed. The progressive amounts of antioxidant added (celastrol) to the voltaic cell allow us to measure the consumed superoxide (at the ring electrode) more precisely than when using only one working electrode in classical cyclovoltammetry [41].

Celastrol is a methide quinone having two hydroxyl groups, Scheme 1, and these may be involved in sequestering superoxide, as clarified through DFT calculation. From the RRDE graph, Figure 4, it is seen that progressive amounts of celastrol decrease but cannot deplete the superoxide content in the voltaic cell. The signal detected at the ring electrode is located at the upper part of the graph, and shows that at the maximum celastrol concentration, 9.0 × 10^−5^ M, there is still superoxide in the voltaic cell.

**Figure 4 ijms-21-09266-f004:**
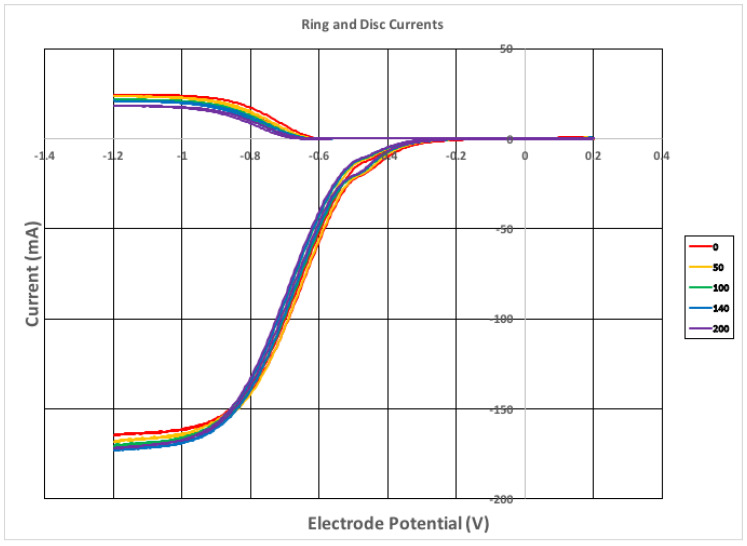
Rotating ring disk electrode (RRDE) voltammograms showing only 5 lines out of 15 for clarity, indicating the volume of aliquots added. The complete set of 15 data is shown in Figure 5.

The collection efficiency is the ratio between current at the ring and the disk for any studied solution, and the slope is the indicator of antioxidant capability. Therefore, celastrol slope, −5.9 × 10^4^, compares well with emodin −6.0 × 10^4^ [42]; it is more active than embelin, −2.8 × 10^4^ [34], and weaker than quercetin, −15.4 × 10^4^ [34] (all measured following the same experimental protocol). Since celastrol is a better scavenger of superoxide than embelin, which has been identified as an inhibitor of the main COVID-19 protease [28], we consider if celastrol might be a better inhibitor.

### 2.3. DFT Antioxidant Properties

Polyphenol action for scavenging radicals is generally based on the capability of radical induced cleavage of a hydroxyl H atom bound to the aromatic ring (option ***a***). The original radical, after binding the H atom, becomes electron paired, while the former polyphenol becomes a newly formed, more stable radical since the unpaired electron relocates within the aromatic ring framework and is less likely to react with other molecules. However, a second possibility of scavenging (option ***b***) is feasible when the ring system captures only the electron directly from the original radical, leaving the polyphenol aromatic hydroxyls unaffected. Very probably, option (***b***) is favored when polyphenol hydroxyls are inaccessible, for instance, due to strong H-bonds. Interestingly, option (***b***) is better suited when a scavenger contains a quinone ring rather than an aromatic system. Our lab has studied the quinones embelin and emodin [34,42], two natural products, and we found that they prefer option (***b***).

Figure 6 and Figure 7 show the approach of superoxide to a hydroxyl belonging to a simplified celastrol model, used to decrease calculation time. Initial coordinates of one celastrol molecule were obtained from our X-ray diffraction study. After DFT geometry optimization, the transfer of only one electron from the superoxide radical to the methide quinone ring is seen, which illustrates option (***b***). Figure 8 and Figure 9 show the equivalent approach through a π-π interaction, with equal results. Results of more extensive calculations are shown in Figure 10 and Figure 11, where the complete molecule of celastrol shows additional reactivity. Thus, after van der Waals π-π interaction by a hydronium H_3_O^+^ cation to the O=C-COH moiety, a proton is captured by the O(carbonyl), with formation of a molecule of water. After elimination of water and O_2_ from Figure 10 structure, the resulting neutral semiquinone can further react with an additional superoxide, see Figure 11, leading to option (***a***) mode of radical scavenging by polyphenols described earlier. Formation of H_2_O_2_ would follow when interacting with an additional proton. The whole process is able to sequester more than one superoxide radical, the first σ or π reacted radical is followed by H_3_O^+^ in Figure 5, and an additional superoxide shown in Figure 6. The outcome of this process is therefore defined by the availability of superoxide and protons, and illustrated in the RRDE section. That is, superoxide radical anions are abundantly provided by the voltaic potential, whereas not completely anhydrous DMSO solvent and/or air environmental humidity can influence the RRDE outcome.

Additional examination of the celastrol antioxidant action through its ring E carboxylic moiety is now described. Geometry optimization of the arrangement containing the initial van der Waals-separated moieties [O(superoxide)---H(carboxylic moiety)] is shown in Figure 12. When the van der Waals-separated probable product [O_2_H---Celastrol-carboxylate] was geometry-optimized the same structure seen in Figure 12 was obtained. Thus, the DFT analysis shows the formation of a stable complex between superoxide and celastrol when interacting through its carboxylic acid moiety. Furthermore, the LUMO shown in Figure 12 suggests that the carboxylic-superoxide complex does not preclude further reactivity at ring A. Therefore, celastrol demonstrates strong antioxidant activity (Figure 7, Figure 9, Figure 10 and Figure 11) through the combination of superoxide scavenging and incorporation of protons (ring A) as well as its capability to stabilize an additional superoxide molecule at the other end of the molecule, ring E, Figure 12.

### 2.4. Docking the Protease COVID-19 3CL^pro^

The SARS 3CL^pro^ protease active site was obtained after downloading coordinates from the PDB database, PDB ID: 6LU7 [43]. The published protein included the inhibitor N3 and was treated with the Discovery Studio “prepare protein” protocol, which provides assignment of CHARMm force field and addition of H atoms. The inhibitor was removed and the binding site was chosen after selecting Cys145 and His41 amino acids, for a sphere of radius 14 Å. Results of docking celastrol (15 poses) were analyzed focusing attention on interactions between S(Cys145) and the methide quinone centroid, which were found in closely related poses 5, 13, and 14. Pose 5 has CDOCKER interaction energy of −41.7 kcal/mol and was selected for further study. Figure 13 displays essential amino acids useful for description of the mechanism. Pose 5 was also treated with Discovery Studio standard dynamic cascade protocol, Figure 14, and results show a marked increase in interactions between celastrol and the active site. Thus, H(His41) H-bond to celastrol carbonyl of 3.184 Å (docking) gets shorter after dynamics (2.181 Å); S(Cys145) directed to the quinoid carbonyl, 4.612 Å (docking), becomes slightly closer after dynamics, 4.559 Å. The dynamic cascade shows O(His164) H bond to H-S(Cys145) of 2.138 Å, suggesting H capture and thiolate formation, for further attack on the celastrol carbonyl by the latter. Thus, capture of H(His41) by the celastrol carbonyl, and O(His164) capture of H-S(Cys145) generating S(Cys145) thiolate, are shown in Figure 15, having the S(thiolate) pointing to the celastrol C(carbonyl), 3.347 Å. In addition, further stabilization of the celastrol-protease complex is provided by Met49, which was initially distant from celastrol in docking pose 5, but after dynamic cascade gets closer to the ring B centroid of celastrol, 4.480 Å, establishing a π interaction, in agreement with ring B capability to interact with nucleophilic S atoms [20]. The 2D interaction display for celastrol after dynamic cascade is shown in Figure 16, while that for Cys145 showing H-bond to O(His164) is depicted in Figure 17.

In conclusion, celastrol is a methide quinone compound obtained from the roots of *Tripterygium wilfordii* and used in traditional Chinese medicine. Celastrol also shows a large variety of interesting biological properties, including inhibition of COVID-19. In this study, we describe celastrol interaction with the active site of the COVID-19 virus main protease. We also include a crystal structure analysis of celastrol and disclose its uncommon antioxidant capability when scavenging the superoxide radical, by using DFT methods. Thus, there are two chemical moieties in celastrol able to scavenge the superoxide radical: the carboxylic framework of ring E, and more importantly, the methide quinone located at ring A. The latter captures the superoxide electron, releasing molecular oxygen. This unusual scavenging of the superoxide radical is supported experimentally by cyclic voltammetry and X-ray diffraction. We observe a correlation between antioxidant property and protease inhibition, through a specific interaction of celastrol with the Cys145 amino acid, assisted by proton transfer at the active site. The final complex formed between celastrol and Cys145 thiolate is further stabilized by a π-bonding between Met49 and the B ring of celastrol.

Therefore, our proposed mechanism for celastrol inhibition, based on molecular dynamics, is shown in Scheme 2 and consists of (1) proton donation to celastrol O(carbonyl) by HN(His41) and (2) cleavage of S-H bond in Cys145 by His164. These first two steps determine formation of intermediate (3). Subsequently, (4) attack of S(thiolate)-Cys145 to celastrol former O(carbonyl) and (5) S(Met49) π bonding to the ring B centroid define the final product that inhibits the protease.

Finally, this study of celastrol was stimulated by its already described anti-SARS-2 biological activity [28], and how its methide-quinone structural features compare with the embelin mechanism on COVID-19 main protease inhibition [27]. Celastrol-COVID-19 investigation is also encouraged because of promising in vivo results from celastrol on a variety of lung related disorders [44]. Our results allow us to correlate the antioxidant property of celastrol, when scavenging the superoxide radical, with its inhibitory profile on the main protease COVID-19 active site. We also describe the inhibitory detailed mechanism of celastrol, based on docking and dynamics molecular mechanics. 

## 3. Materials and Methods

### 3.1. Chemicals

Celastrol (Cayman, Ann Arbor, MI, USA). Dimethyl sulfoxide (DMSO, anhydrous, ≥99.9%), tert-butyl ammonium bromide (TBMA), [(2,2-dimethyl-6,6,7,7,8,8,8-heptafluoro-3,5-octanedionato)silver(I)], (Sigma-Aldrich, St. Louis, MO, USA)

### 3.2. Equipment

Hydrodynamic voltammetry at a rotating ring-disk electrode (RRDE) was carried out using the Pine Research (Pine Research, Durham, NC, USA) WaveDriver 20 bipotentiostat with the Modulated Speed Electrode Rotator. The working electrode is the AFE6R2 gold disk and gold ring rotator tip (Pine Research, Durham, NC, USA), combined with a coiled platinum wire counter electrode and a reference electrode consisting of an AgCl coated silver wire immersed in 0.1 M TBAB in dry DMSO in a fritted glass tube. The electrodes were placed in a five-neck electrochemical cell together with means for either bubbling or blanketing the solution with gas. Voltammograms were collected using Aftermath software provided by Pine Research. Careful cleaning of the electrodes was performed by polishing with 0.05 µm alumina-particle suspension, Allied High Tech Products, Inc, Rancho Dominguez, CA, USA, on a moistened polishing microcloth to eliminate potential film formation [45].

### 3.3. RRDE Study

Stock solutions of 0.022 M celastrol in anhydrous DMSO were used in trials. For the experiment, a solution of 0.1 M TBAB in anhydrous DMSO was bubbled for 5 min with a dry O_2_/N_2_ (35%/65%) gas mixture to establish the dissolved oxygen level in the electrochemical cell. The Au disk electrode was then rotated at 1000 rpm while the disk was swept from 0.2 V to −1.2 Volts and the ring was held constant at 0 Volts; the disk voltage sweep rate was set to 25 mV/s. The molecular oxygen reduction peak (O_2_ + e^−^→O_2_^−•^) was observed at the disk electrode at −0.6 volts; the oxidation current (O_2_^−•^→O_2_ + e^−^) is observed at the ring electrode. An initial blank was run on this solution, and the ratio of the peak ring current to disk current was calculated as the “collection efficiency” in the absence of antioxidant. Next, an aliquot of the antioxidant was added, the solution bubbled with the gas mixture for 5 min, and the CV rerecorded. Again, the reduction and oxidation peaks were measured and the collection efficiency was calculated. Any decrease in the collection efficiency was due to the amount of superoxide removed by the antioxidant.

### 3.4. Diffraction Study

The molecular structure of celastrol was determined at low temperature (125 K), and deposited at the CSD database, with Deposition Number 2036458. Crystal Data are reported in Table 1. Single crystal X-ray diffraction data were collected using a Bruker APEX II diffractometer using λ = Cu Kα 1.54178 Å radiation. Data integration scaling was done using Bruker software. Structure solution and refinement was done using SHELX-L [46]. Heavy (non-hydrogen) atoms were refined with their anisotropic displacement parameters. Hydrogen atoms were placed in calculated positions and refined with a riding model on the atoms they were attached to.

### 3.5. Computational Study

Calculations were performed using software from Biovia (San Diego, CA, USA). Density functional theory (DFT) code DMol^3^ was applied to calculate energy, geometry, and frequencies implemented in Materials Studio 7.0 [47]. We employed the double numerical polarized (DNP) basis set that included all the occupied atomic orbitals plus a second set of valence atomic orbitals, and polarized d-valence orbitals [48]; the correlation generalized gradient approximation (GGA) was applied including Becke exchange [49] and Perdew correlation (PBE) [50]. All electrons were treated explicitly, and the real space cutoff of 5 Å was imposed for numerical integration of the Hamiltonian matrix elements. The self-consistent field convergence criterion was set to the root mean square change in the electronic density to be less than 10^−6^ electron/Å^3^. The convergence criteria applied during geometry optimization were 2.72 × 10^−4^ eV for energy and 0.054 eV/Å for force. Calculations include DMSO solvent effect for proper comparison with RRDE results. Docking and molecular dynamic studies were performed with the CDOCKER package in Discovery Studio 2020 version [51]. For molecular dynamics, we used default conditions.

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
