# Peer review of "Interrelated Mechanism by Which the Methide Quinone Celastrol, Obtained from the Roots of Tripterygium wilfordii, Inhibits Main Protease 3CLpro of COVID-19 and Acts as Superoxide Radical Scavenger"

_ijms, 2020, doi:10.3390/ijms21239266_

Round 1

Reviewer 1 Report

The authors found that celastrol, as the active compound of Tripterygium wilfordii, is an inhibitory factor at the main active site of the Covid-19 protease. Authors also describe the inhibitory detailed mechanism of celastrol, based on docking and dynamics molecular mechanics. Their finding is interesting, but the work requires a few corrections.

  • The title is unclear and needs to be changed. The natural product is Tripterygium wilfordii. Celastrol is active compound of Tripterygium wilfordii
  • Authors should provide more information about Tripterygium wilfordii.It is known that, despite its impressive therapeutic properties, the safety of Tripterygium wilfordii is not fully understood. It is suspected that plant extracts may cause intestinal toxicity, reproductive toxicity, hepatotoxicity, nephrotoxicity, hematotoxicity, dermal toxicity and other damage.
  • Authors should also provide more information about the active compound, celastrol (e.g., conclusion or discussion). They should focus on the toxicological profile and limitations of celastrol, namely solubility, bioavailability, and maybe dosing issues that still limit its further clinical use and utility.

Author Response

The title is unclear and needs to be changed. The natural product is Tripterygium wilfordii. Celastrol is active compound of Tripterygium wilfordii

Response: The title has been changed accordingly

Authors should provide more information about Tripterygium wilfordii.It is known that, despite its impressive therapeutic properties, the safety of Tripterygium wilfordii is not fully understood. It is suspected that plant extracts may cause intestinal toxicity, reproductive toxicity, hepatotoxicity, nephrotoxicity, hematotoxicity, dermal toxicity and other damage.

Response: done, page 2,  lines 32-41

Authors should also provide more information about the active compound, celastrol (e.g., conclusion or discussion). They should focus on the toxicological profile and limitations of celastrol, namely solubility, bioavailability, and maybe dosing issues that still limit its further clinical use and utility.

Response: In part at page 2, lines 32-41. Additional conclusions are now included, pages 23-2, lines 296-316.

Reviewer 2 Report

The authors have analyzed the compound celastrol, its antioxidant properties and the inhibition of Covid-19 main protease by antioxidant measurements and computational study.

A lot of different results are put together. I felt that such a manuscript should have stressed out their findings in the conclusion. Moreover, the abstract seems to be cut off and is missing major findings. 

Author Response

A lot of different results are put together. I felt that such a manuscript should have stressed out their findings in the conclusion. Moreover, the abstract seems to be cut off and is missing major findings. 

Response: Abstract has been more elaborated accordingly. Conclusions are also more expanded, pages 23-24, lines 296-316